



# Variability and trends of the surface solar spectral ultraviolet irradiance in Italy: a possible influence of lower and upper stratospheric ozone trends

Ilias Fountoulakis[1,a], Henri Diémoz[1], Anna Maria Siani[2], Alcide di Sarra[3], Daniela Meloni[3], Damiano M. Sferlazzo[4]

[1]Aosta Valley Regional Environmental Protection Agency (ARPA), 11020 Saint-Christophe, Italy
[2]Physics Department, Sapienza Università di Roma, 00185 Rome, Italy
[3]ENEA, Laboratory for Earth Observations and Analyses, 00123, S. Maria di Galeria, Rome, Italy
[4]ENEA, Laboratory for Earth Observations and Analyses, 92010, Lampedusa, Italy
[a]Now at: Institute for Astronomy, Astrophysics, Space Applications and Remote Sensing, National Observatory of Athens (IAASARS/NOA), 15236 Athens, Greece

*Correspondence to*: Henri Diémoz (h.diemoz@arpa.vda.it)

**Abstract.** In this study the short- and long-term variability of the surface spectral solar ultraviolet (UV) irradiance are investigated over Italy using high quality ground based measurements from three sites located at quite different environmental conditions, and covering the full latitudinal extent of the Italian territory: Aosta (45.7° N, 7.4° E, 570 m a.s.l.), Rome (41.9° N, 12.5° E, 75 m a.s.l.), and Lampedusa (35.5° N, 12.6° E, 50 m a.s.l.). The variability of the irradiances at 307.5 nm, 324 nm, and of the ratio between the 307.5 nm and the 324 nm irradiances were investigated with respect to the corresponding variability in total ozone and the geopotential height at 250 hPa (GPH). The study was performed for two periods: 2006 – 2020 for all stations, and 1996 – 2020 only for Rome. A statistically significant correlation between the GPH and total ozone monthly anomalies was found for all stations and all seasons of the year. A corresponding statistically significant correlation was also found in most cases between the GPH and the 307.5 nm irradiance monthly anomalies. The correlation between GPH anomalies at different sites was statistically significant, possibly explaining the strong and significant correlation between the total ozone monthly anomalies at the three sites. A statistically significant decrease of total ozone, of ~0.1%/year was found for Rome for the period 1996 – 2020, which however did not induce increasing trends in irradiance at 307.5 nm (neither increasing trends in the ratio between the 307.5 nm and the 324 nm irradiances) at SZA = 67°. Further analyses revealed positive trends in the ratio and the 307.5 nm irradiance at smaller solar zenith angles (SZA), which can be attributed to the fact that total ozone decrease is driven by a decrease in the lower stratosphere while upper stratospheric ozone increases, and the effect of changes of upper stratospheric ozone becoming disproportionately larger for increasing SZA. It was also showed that long-term changes in total ozone follow changes in GPH, which is an additional indication that negative trends in total ozone are mainly driven by changes in lower stratospheric ozone. An anti-correlation between the GPH long-term changes and total ozone was also evident for all stations in 2006 – 2020. Positive trends in UV irradiance for this latter period which were possibly driven by changes in clouds and/or aerosols were found for Rome and Aosta. This study clearly points out the



significance of dynamical processes which take place in the troposphere for the variability of total ozone and surface solar UV irradiance.

## 1 Introduction

The amount of solar ultraviolet radiation (UVR) reaching the Earth's surface is an important environmental, ecological and atmospheric parameter to be measured and studied. The relationship between UVR and biological effects has been well established. Exposure to UVR is vital for many living organisms, including humans, however, overexposure may produce detrimental effects in humans, animal and plants (Paul and Gwynn-Jones, 2003; Caldwell et al., 1998; Bornman et al., 2019; Calkins and Thordardottir, 1980; Williamson et al., 2019; Häder et al., 1998). Photons with wavelengths below 290 nm are absorbed in the higher atmosphere, mainly by molecular oxygen ($O_2$) and ozone ($O_3$) and practically do not reach the Earth's surface. Most of the ultraviolet-B (UVB) irradiance (280 – 315 nm) is also absorbed by ozone in the stratosphere (Bais et al., 1993; Griggs, 1968; Inn & Tanaka, 1953). Absorption by stratospheric ozone dominates on scattering by molecules and aerosols in the UV-B spectral range, while scattering plays a relatively larger role than ozone absorption in the UV-A.

In the 1980s and 1990s, the increase of anthropogenic emissions of ozone depleting substances (ODS) enhanced the chemical destruction of ozone in the stratosphere mainly over Arctic and Antarctica, leading to reduced stratospheric ozone even at mid-latitudes of the Southern and the Northern hemispheres (Solomon et al., 1986; McConnell et al., 1992), and subsequently to higher levels of UVB radiation at the Earth's surface (Madronich et al., 1998; Kerr and McElroy, 1993; Zerefos et al., 1995). Emissions of ODS were regulated after the adoption of the Montreal Protocol in 1987 and subsequent amendments and adjustments, and since the mid-1990s the reduction of stratospheric ozone has decelerated. The first signs of recovery are now evident over higher latitudes (Weber et al., 2018; Solomon et al., 2016). Recent studies show that recovering ozone over Antarctica resulted on decreasing UVB radiation (Bernhard and Stierle, 2020). Over the Arctic, many studies report negative, but in most cases not significant, trends of UVB radiation in spring (Eleftheratos et al., 2015; Lakkala et al., 2017; Svendby et al., 2018). Due to the successful implementation of the Montreal protocol the world avoided extremely high levels of solar UVB radiation which would have been detrimental for the viability of ecosystems, as well as for human health (Morgenstern et al., 2008; Newman and McKenzie, 2011; McKenzie et al., 2019).

However, the future evolution of total ozone levels, and subsequently of UVB radiation is still uncertain. Although decreasing ODS since mid-1990s led to increasing ozone in the upper stratosphere since mid-1990s (Steinbrecht et al., 2017; Sofieva et al., 2017), lower stratospheric ozone in the Northern mid-latitudes has been continuously decreasing, offsetting the increase occurring in the upper stratosphere (Wargan et al., 2018; Ball et al., 2018). Changes in lower stratospheric ozone appear to have a strong spatial and seasonal variability, and the processes that drive them are not clear yet (Szeląg et al., 2020). Furthermore, dynamical phenomena occasionally (once every few years) favor extensive destruction of Arctic stratospheric ozone in early spring (Dameris et al., 2021; Manney et al., 2011; Pommereau et al., 2018; Varotsos et al., 2012; Wohltmann et al., 2020), leading to extremely low ozone over Northern hemisphere high and mid-latitudes (due to the transport of poor-





ozone air masses from the poles towards mid-latitudes), and subsequently to very high levels of solar UVB radiation at the Earth's surface (Bernhard et al., 2020; Petkov et al., 2014). In addition, low ozone episodes (Siani et al., 2002) not related to ozone over the poles can be experienced. These episodes are characterized by significant ozone decreases over limited geographical regions and are associated with synoptic weather systems. An increasing frequency of the occurrence of such phenomena in the future could strongly affect the ozone levels in the atmosphere, and subsequently the terrestrial levels of

UVB. As discussed in several studies (e.g., Dobson et al., 1946; Reed, 1950; Hoinka et al., 1996; Varotsos et al., 2004; Angell and Korshover, 1964; Steinbrecht et al., 1998; Vaughan and Price, 1991), changes in tropopause altitude are linked to inverse changes in the amount of total ozone. Thus, increasing altitude of the tropopause due to climate change related warming of the troposphere would induce negative trends in lower stratospheric ozone, and subsequently positive trends in UVB.

Over many mid-latitude stations of the Northern hemisphere, changes in aerosols and clouds – and not ozone - have been

found to be the main drivers of the long-term changes of the UVB and the UVA irradiance (Chubarova et al., 2020; De Bock et al., 2014; Fitzka et al., 2012; Fountoulakis et al., 2016; Fountoulakis et al., 2018; Hooke et al., 2017; Zhang et al., 2019). Aerosols and clouds also play a significant role in the short-term variability of UVA and UVB irradiance (e.g., Mateos et al., 2015; Kazadzis et al., 2009; di Sarra et al., 2008; Mateos et al., 2011). Two different studies (Fragkos et al., 2016), and Raptis et al., 2021) report moderate or even low erythemal irradiance (McKinlay and Diffey, 1987) during extremely low ozone

events, because of remarkably high aerosol load. The trends reported in different studies for the period between mid-1990s to present vary significantly even within a few hundreds of kilometres. Fountoulakis et al., (2020b) reported an average increase of 5% per decade in the 307.5 nm irradiance for Uccle, Belgium in 1996 – 2017, and for the same period an average decrease of 7% per decade for Reading, UK, which is less than 400 km from Uccle. Other, independent studies report similar results for the two stations. Hooke et al. (2017) reported an 8% decrease in erythemal doses for Chilton, UK (located a few kilometers

from Reading) in 1991 - 2015, while De Bock et al., (2014) and Pandey et al., (2016) reported positive trends in Uccle (in 1991 – 2013 and 1995 – 2014 respectively).

The UV irradiance spatial variability in Italy is very large, mainly due to the country's long latitudinal extent and complex topography (Meloni et al., 2000). Thus, atmospheric parameters may affect UVB and UVA irradiance in a different way at different locations (e.g., Fountoulakis et al., 2020). Recent studies have shown that atmospheric parameters which affect

significantly solar UV radiation, such as clouds (Manara et al., 2016; Manara et al., 2015; Mateos et al., 2011; Pfeifroth et al., 2018) and aerosol (Orza and Perrone, 2015; Rizza et al., 2019; Di Ianni et al., 2018; Putaud et al., 2014; Masiol et al., 2017; di Sarra et al., 2008) have changed in Italy in the last decades, either on a regional or on a country scale. Although in Italy long-term continuous solar spectral UV (Diémoz et al., 2011) and total ozone measurements (Siani et al., 2018) are available from different stations across the country, they have never been used to study the long-term changes of UV irradiance at the

surface, and how they are affected by changes in total ozone. In this study, the long-term datasets of high-quality spectral UV and total ozone measurements of three Italian sites (Aosta, Rome, and Lampedusa), located at quite different latitudes and environmental conditions were used in order to study the changes of solar UV irradiance, and the extent at which they are



driven by changes in total ozone. Furthermore, there was an effort to investigate whether, and to what extent changes in synoptical atmospheric circulation affect the surface solar UV irradiance short- (e.g., yearly) and long-term variability.

The paper is structured as follows. In Sect. 2 the data and methods used for the study are described. In Sect. 3.1 the possible links between the short-term variability of UV irradiance and atmospheric dynamics are studied. In Sect. 3.2 the trends of spectral UV irradiance and total ozone for the three stations are analyzed and discussed with respect to their main drivers. Finally, in Sect. 4 the main findings and the conclusions of the study are summarized.

## 2 Data and methodology

Long series of high-quality spectral UV measurements are available from three Italian stations, providing a complete latitudinal coverage of Italy (Fig. 1). The Aosta monitoring station at the facilities of Regional Environmental Protection Agency of the Aosta Valley (ARPA Valle d'Aosta) (45.7° N, 7.4° E, 570 m a.s.l.) is a semi-rural site (at Aosta–Saint-Christophe), in the North-Western Alps. The site of the Physics Department of Sapienza University of Rome (41.9° N, 12.5° E, 75 m a.s.l.) is an urban site at a distance of about 25 kilometres from the Tyrrhenian Sea. The Lampedusa Station for Climate Observations of

the Italian Agency for the New Technologies, Energy and Sustainable Economic Development (ENEA), located on the island of Lampedusa (35.5° N, 12.6° E, 50 m a.s.l.) is a background remote (island) site.

At Rome the total column of ozone has been measured by a single monochromator Brewer (MkIV type) with serial number 67 (Brewer#067) since 1992 (Siani et al., 2018). Measurements of the spectral irradiance have been performed by the same instrument since 1996, at wavelengths 290 – 325 nm with a step of 0.5 nm and a resolution of ~0.6 nm (Casale et al., 2000).

The world travelling reference standard Brewer (Brewer#017) (maintained by the International Ozone Services Inc.; https://www.io3.ca/index.php) transfers the calibration from the reference triad maintained by Environment and Climate Change Canada (Fioletov et al., 2005; Zhao et al., 2020) to field instruments. Hereafter the travelling reference standard Brewer is referred as IOS standard. Intercomparisons between Brewer#067 and the IOS standard are performed on an annual or biennial basis ensuring the optimal quality of total ozone measurements. As discussed in the following, regular inter-

comparisons between the IOS standard and the Brewer spectrophotometers at Lampedusa and Aosta ensure the consistency between the total ozone datasets recorded at the three stations. Inter-comparisons with the UV reference standard Quality assurance of spectral ultraviolet measurements in Europe through the development of a transportable unit (QASUME; Hülsen et al., 2016; Gröbner and Sperfeld, 2005) (hereafter referred as QASUME) in 2003 – 2008 and with IOS standard thereafter, performed every 2-3 years, ensure the high quality of the spectral UV measurements. The consistency between the UV spectra

measured by QASUME and the IOS standard is ensured by the common participation of both instruments in the inter-comparison campaigns on an annual or biennial basis (e.g., Redondas et al., 2018; Redondas & Rodriguez-Franco, 2015). For the present study measurements of total ozone and spectral UV irradiance for 1996-2020 have been used. Note that spectral UV measurements for the first six months of 2018 were not used for Rome because operational problems induced increased uncertainty in the measurements.



At Lampedusa measurements of total ozone and spectral UV irradiance have been performed since 1997 by a double monochromator Brewer (MkIII type) with serial number 123 (Brewer#123) (di Sarra et al., 2002). Solar spectra are measured in the range 286 – 363 nm with a step of 0.5 nm and a resolution of ~ 0.55 nm. Calibrations of spectral UV measurements are performed 4-5 times per year with a field calibrator which uses 1000 W FEL lamps traceable to NIST (di Sarra et al., 2008). Regular inter-comparisons (every 2 – 4 years since 1998) with the IOS standard ensure the good quality of total ozone

measurements. The Lampedusa spectral UV dataset has been recently subjected to QA/QC and homogenized for the period 2003 – 2020. For the present study measurements of the total ozone and spectral UV irradiance have been used for the period for which high quality measurements are available for all three sites (i.e., 2006 – 2020).

At Aosta, spectral UV-visible measurements are performed by a double monochromator Bentham DTMc300 with serial number 5541 (Bentham5541) in the range 290 – 500 nm with a step of 0.25 nm and a resolution of ~0.5 nm. Spectral UV

measurements began in 2004. Since 2006 inter-comparison with QASUME is performed on an annual or biennial basis ensuring the high quality of the measurements. Recently the spectral UV dataset of Aosta was re-evaluated and homogenized for the period 2006 – 2020 (Fountoulakis et al., 2020a). Measurements of the total ozone are available since 2007 by a single monochromator Brewer (MkIV type) with serial number 66 (Brewer#066). The quality of total ozone measurements is ensured by inter-comparisons with the IOS standard, which have been performed since 2007 on a biennial basis. Total ozone and

spectral UV measurements for Aosta have been analysed for 2007 – 2020 and 2006-2020, respectively.

The good quality of the total ozone and spectral UV measurements is further ensured by performing stability checks and applying a number of quality control/quality assurance procedures on a regular (daily, weekly, or monthly basis) at each of the three sites described above (Fountoulakis et al., 2020b; Ialongo et al., 2010). The accuracy of total ozone columns retrieved by measuring irradiances of direct sunlight radiation (DS mode) (Kerr, 2010) by well-maintained and calibrated Brewer

spectrophotometers, such as those used in the present study, is of the order of 1% (Vanicek, 2006). The consistency between measurements from different instruments (if they are well maintained and calibrated) is also ~1% (Redondas et al., 2018). The standard uncertainty for spectral UV measurements at wavelengths longer than 305 nm from well maintained and calibrated Brewer spectrophotometers is of the order of 5% (Garane et al., 2006). The agreement between synchronous spectra from well maintained and similarly calibrated spectrophotometers is also of the order of 5% (standard deviation of the differences) (Bais

et al., 2001). For wavelengths longer than 305 nm and SZAs below 75° the standard uncertainty in the spectral UV measurements of Bentham5541 is smaller than 2.5%. A detailed description of the uncertainties of the spectral UV measurements performed in Aosta can be found in Fountoulakis et al., (2020a).

Spectral measurements at the wavelengths 307.5 nm (306.5 – 308.5 nm average) and 324 nm (323 – 325 nm average) have been analysed for all three sites. Irradiance at 307.5 nm is strongly affected by total ozone, while the irradiance at 324 nm is

weakly affected by ozone. Although ozone has a stronger effect on wavelengths shorter than 307.5 nm, the particular wavelength was chosen because it is less affected by noise and spectral straylight (relative to shorter wavelengths). For each spectrum, the ratio between the irradiance at the two wavelengths (307.5 nm and 324 nm) was also calculated and will be herein referred as 307.5/324 nm ratio. Long-term changes of the ratio should be strongly correlated with changes in total ozone.





For each day, all spectra measured within +/- 2° around the solar zenith angles (SZA) of 67° and 45° were interpolated to the
central SZAs using the empirical relationship proposed by Fountoulakis et al. (2016), and then averaged to calculate the
irradiance. The SZA of 67° is reached throughout the whole year at all three sites, while the SZA of 45° is reached in the period
April – September. Monthly averages of the irradiances and the ratio were calculated for months for which measurements were
available for at least 15 days. Daily averages of the total ozone have been used for the calculation of monthly averages, again
when measurements for at least 15 days were available. For all three stations, analyses were performed for the period
September 2006 – August 2020 (14 years), i.e., the longest period of overlapping measurements. For Rome, analyses were
additionally performed for the period September 1996 – August 2020 (24 years). In all cases, monthly climatological averages
of each quantity were calculated for the whole available period (2006 – 2020 or 1996 – 2020). Monthly anomalies were
calculated by subtracting the monthly climatological averages from the monthly average values. Calculation of trends was
performed by applying a least square linear fit to the monthly anomalies and statistical significance was in all cases estimated
by applying the Mann-Kendal test.

In addition to the ground-based measurements, re-analysis products from Modern-Era Retrospective analysis for Research and
Applications version 2 (MERRA-2) have been used, mainly for the investigation of changes in atmospheric conditions, and
ozone at different atmospheric levels. All MERRA-2 products used in the present study were provided in a grid resolution of
0.5°x0.625° (latitude x longitude) and have been downloaded from the Giovanni platform maintained by National Aeronautics
and Space Administration (NASA) (https://giovanni.gsfc.nasa.gov/giovanni/). Monthly average total ozone (GMAO, 2015a)
for 1996 – 2020 and Geopotential Heights (GPH) at 250 hPa and 850 hPa (GMAO, 2015b) have been downloaded and analysed
for southern Europe. The Geopotential Height (GPH) was also linearly interpolated to the co-ordinates of Aosta, Rome and
Lampedusa in order to study its variability over these sites.  Monthly average ozone mixing ratio (GMAO, 2015b) from
MERRA-2, at different pressure levels between 150 and 3 hPa for 1996 – 2020 has been interpolated to the coordinates of the
station of Rome. The version 7 Aqua/AIRS L3 Monthly Standard Physical Retrieval (AIRS-only) tropopause height product
with 1° x 1° resolution (AIRS3STM) (AIRS project, 2019) was also interpolated to the coordinates of the three stations for the
period 2006 – 2020. The AIRS3STM was also downloaded from the same platform. As shown later, positive/negative
anomalies in GPH at 250 hPa generally coincide with positive/negative anomalies of the tropopause altitude and
positive/negative anomalies in GPH at 850 hPa. According to the studies of Varotsos et al., (2004) and Steinbrecht et al.,
(1998), positive trends in the height of the tropopause explain 25 – 30% of the reduction of total ozone over Athens in 1984 –
2002, and Hohenpeissenberg in 1967 – 1997 respectively.

### 3 Short- and long-term variability of UV irradiance

### 3.1 Short-term variability of UV irradiance and the role of changes in GPH

In this section we investigate whether there is a correlation between the monthly anomalies of the GPH at 250 hPa (hereafter
named as GPH unless something different is specified) and the corresponding anomalies of UV irradiance (at 307.5 nm, 324



nm) and the 307.5/324 nm ratio. Here, and in the following sections, the Pearson (i.e., linear) correlation coefficients are reported. For all cases that a strong correlation was found, it was optimally described by a linear fit. The investigation was performed for all three stations for the period 2006 – 2020. For Rome, the analyses were also performed for the period 1996 – 2020.

Fig. 2 shows the sequence of the correlation analysis among the sites for the different atmospheric parameters. As shown in Fig. 2, the GPH at Aosta and Rome varies in a similar way despite the distance between them and the fact that the two sites are at quite different environments. The correlation coefficient for GPH at the two sites (panel a) is ~0.9 and indicates that to a wide extent the two sites are affected by the same synoptical systems. This strong correlation is also found for the anomalies of total ozone (panel m) and subsequently for the anomalies of the 307.5 nm irradiance (panel d) and the 307.5/324 nm ratio

(panel j). The correlation for the 324 nm irradiance (panel g) is statistically significant but weak (~0.26). The corresponding correlation between the same variables (with the exception of the 324 nm irradiance) at Rome and Lampedusa was again significant but weaker than for Rome and Aosta, which however was expected since the latitudinal distance between Rome and Lampedusa (6.4°) is longer than the distance between Rome and Aosta (3.8°). Part of the differences could be also due to the fact that dust outbreaks play a more significant role on the variability of surface UVR over Lampedusa than in Rome and

Aosta (e.g., Meloni et al., 2008). A weak but statistically significant correlation between GPH (~0.5) and total ozone (~0.4) in Aosta and Lampedusa was found. However, in this latter case the correlation found for total ozone and GPH is not reflected in the levels of UV irradiance. Further analyses showed that for GPH and total ozone the correlation between different sites was generally stronger in winter and weaker in summer. Even for summer, however, the correlation between the aforementioned parameters for the pairs Aosta – Rome and Rome – Lampedusa is statistically significant (in the case of total

ozone the correlation is also significant for the pair Lampedusa – Aosta). What is interesting is that the strongest correlation for the 307.5 nm irradiance and the 307.5/324 nm ratio for all sites was found for summer, which shows that the levels of the 307.5 nm irradiance over all three sites vary in a similar way, mainly because total ozone also varies in a very similar way.

In Table 1 the correlation coefficients between GPH and the UV irradiance at 307.5 nm and 324 nm (for SZAs 67° and 45°), GPH and the 307.5/324 nm ratio (for SZA = 67° and SZA = 45°), and GPH and total ozone are presented. The monthly

anomalies for each of the four seasons of the year (December, January, February for winter, March, April, May for spring, June, July, August for summer, and September, October, November for autumn) were used for the calculation of the correlation coefficients. Numbers in bold denote statistical significance (Mann – Kendall method was used to determine statistical significance) at the 95% confidence level (hereafter the statistical significance at the 95% confidence level is referred as statistical significance).

Variations in the GPH at 250 hPa are strongly correlated with variations in the tropopause altitude, as well as with variations in the GPH at 850 hPa (i.e., near the surface) (see Appendix I). Thus, the results presented in Table 1 partially confirm the findings of previous studies reporting correlation between tropospheric dynamics variability and total ozone, since a strong and statistically significant anti-correlation between total ozone and GPH was also found for all Italian stations and all seasons. The correlation is slightly lower in summer relative to other seasons (especially in Aosta). A possible explanation is that the





relatively stable synoptical conditions in summer lead to much smaller variability in GPH relative to other seasons of the year, which in turn possibly has a weaker effect on total ozone. As discussed in the following sections, changes in GPH also play a significant role in the long-term changes of total ozone over Italy.

The anti-correlation between total ozone and GPH was associated with, in most cases, a statistically significant correlation between GPH and the 307.5/324 nm ratio. The correlation between the GPH and the ratio was generally stronger for Aosta

than for Lampedusa and Rome. This is possibly because variations of the GPH are linked to larger variations of the total ozone at higher latitudes, given that the amount and the variability of total ozone increase with increasing latitude. The correlation between GPH and the 307.5 nm irradiance was also statistically significant in most cases. In some cases (e.g., Aosta in autumn) the correlation between GPH and the 307.5/324 nm ratio is much stronger than the correlation between GPH and the 307.5 nm irradiance. As already discussed, the variability of the ratio was mainly determined by the variability of total ozone, while the

307.5 nm irradiance was also affected by the variations of factors such as clouds and most types of aerosols which have a relatively flat spectral effect in the range 307.5 – 324 nm.

It is interesting to note that in addition to the significant correlation between GPH and the 307.5 nm irradiance, strong and statistically significant correlation was found between GPH and the 324 nm irradiance for Rome. For 45° the correlation was significant for spring and summer. For 67° the correlation was significant for all seasons for 2006 – 2020, and for all seasons

except summer for 1996 – 2020. The correlation coefficients were generally larger for 2006 – 2020 relative to 1996 – 2020. Statistically significant correlation between the 324 nm irradiance and the GPH has been found for Lampedusa in winter and for Aosta in autumn. The strong link between GPH and the 324 nm irradiance may be related to changes in aerosols and clouds, associated with changes in synoptical meteorological conditions. Aerosol and clouds, in their turn, affect the levels of both UVB and UVA irradiance levels. Changes in GPH at 250 hPa were strongly anti-correlated to changes in atmospheric pressure

near the surface (see Fig. S.1 in the supplement), which were strongly linked to changes in cloudiness and wind patterns (both of which also affect aerosol load).

The results presented in Table 1 show clearly that total ozone and surface UVR are affected not only by dynamical processes that take place in the stratosphere (e.g., Monge-Sanz et al., 2003; Neu et al., 2014; Weber et al., 2011), but they can be also affected significantly by processes that take place in the troposphere.

**3.2 Long-term variability of spectral UV irradiance and total ozone**

**3.2.1 Long-term variability in the period 2006 – 2020**

In this section the results of the analysis of the long-term changes for the same quantities as those discussed in Sect. 3 are presented and discussed for the period September 2006 – August 2020, during which, measurements were available for all three stations: Aosta, Rome, and Lampedusa. The estimated trends of the irradiances at 307.5 nm and 324 nm, and the

307.5/324 nm ratio at SZA = 67° are presented in Fig. 3, while the corresponding trends for SZA = 45° are presented in Fig.





4. The trends were also calculated separately and are presented for each month of the year. The estimated trends for total ozone (from ground-based measurements) and GPH are presented in Fig. 5.

Statistically significant change (of 0.15%/year) was found only for the 307.5 nm irradiance in July at Aosta (Fig. 3a). For the same month neither the 307.5/324 nm ratio nor the 324 nm irradiance changed significantly, although they increased with an

average rate of ~ 0.7%/year. It is possible here that the natural variability masks the presence of trends. The overall increase of the 307.5 irradiance for 2006 – 2020 is ~2%, and although it is statistically significant, it is less than the standard uncertainty in the measurements. Thus, this result should be treated with caution. In November an increase in total ozone over Aosta, of 0.7%/year was found, which again coincides with a decrease of the GPH, which however is not statistically significant. The 307.5 nm irradiance and the 307.5/324 nm ratio also decreased in the same month, but their decrease was again not statistically

significant.

At the SZA of 45° the 324 nm the irradiance in August increased over Aosta by 0.6%/year. An increase of similar magnitude, but not statistically significant, was found for the 307.5 nm irradiance, while the 307.5/324 nm ratio remained relatively stable. In the same month total ozone did not change, which justifies the absence of significant trend in the ratio. However, the GPH increased significantly by 0.07%/year, which may be related with changes in cloudiness and/or aerosols (e.g., Manara et al.,

2016)). Since changes in aerosols and cloudiness have a stronger effect on the direct component of the solar irradiance (scattered irradiance in the UV is mainly redistributed rather than backscattered by aerosols and clouds) the changes in UV are more pronounced for 45° relative to 67° because direct component has a more significant contribution at smaller SZA.

At Lampedusa neither the UV irradiance nor the ratio changed significantly for both SZAs (45° and 67°). Total ozone also did not change at Lampedusa for none of the twelve months. The statistically significant increase in GPH in September at

Lampedusa does not coincide with any statistically significant change of ozone or UV irradiance.

The UV irradiance in April at Rome did not change significantly at SZA = 67°, but increased significantly for both, 307.5 nm and 324 nm, by ~1%/year at SZA = 45°. The 307.5/324 nm ratio also increased, but the increase was not statistically significant. In the same month total ozone decreased and GPH increased, but again, neither of these latter changes was significant. These results show that changes in total ozone are not the dominant driver of changes in UV irradiance in 2006 – 2020, and that other

factors such as aerosols and clouds play a more important role.

**3.2.1 Long-term variability in the period 1996 – 2020**

In Fig. 6 the monthly anomalies of the 307.5 nm irradiance, the 324 nm irradiance, the 307.5/324 nm ratio, and the total ozone are presented for Rome for the period September 1996 – August 2020 at the SZA of 67.5°. Moving averages and the linear trends for the four quantities are also presented in the same figure. The calculated trends and the corresponding p-values are

presented in Table 2. A negative, statistically significant trend of -0.1%/year was found for total ozone, corresponding to a decrease of ~2.4% during the 24-year period. Although total ozone decreases, neither the 307.5 nm irradiance, nor the 307.5/324 nm ratio increase.



The trends for each month were calculated for the 24-year period and are presented in Fig. 7. Analyses were performed for both SZAs, 67.5° and 45°. In addition to the trends of spectral irradiance, the results for the 307.5/324 nm ratio are presented.

The long-term changes of GPH and total ozone (measured by Brewer#067) were also investigated and are presented in Fig. 8. For the SZA of 67° the irradiance at both wavelengths did not change significantly for any of the 12 months of the year. The ratio decreased significantly only in March, on average by -0.2%/year. In general, significant changes in total ozone did not correspond to significant changes in the 307.5/324 nm ratio as expected. Total ozone decreased significantly in April (with an average rate of -0.4%/year), September (with an average rate of -0.2%/year), and October (with an average rate of -

0.15%/year). However, for the SZA of 45° the 307.5/324 nm ratio increased significantly in April (by ~0.35%/year), August and September (for both months by 0.2%/year), inversely following the negative total ozone trends. Unfortunately, measurements for SZA=45° are available only for the period April – September at Rome due to the geographical position of the station.

It can be also perceived from Fig. 8 that trends in total ozone inversely follow the trends of GPH, which is a strong indication

that trends in total ozone are related with dynamical changes in troposphere and the lower stratosphere, which in turn shows that at least part of the trends is due to reduction of ozone at the lower stratosphere. Many recent studies show that at the mid-latitudes of the northern hemisphere ozone increases at the higher stratosphere and decreases at the lower stratosphere (Sofieva et al., 2017; Eleftheratos et al., 2020; Ball et al., 2019; Staehelin et al., 2001). Thus, an explanation of the results presented in Figures 6, 7 and 8 could be that the negative trends in lower stratospheric ozone dominate over positive trends in higher

stratospheric ozone at Rome, resulting to an overall decrease in total ozone. As the SZA increases the role of ozone at the middle and upper atmosphere becomes more important regarding the attenuation of the UV-B irradiance relative to ozone at the lower stratosphere. Qualitative analysis of the monthly average ozone mixing ratio at different pressure levels between 100 and 3 hPa for the period of study over Rome (see Fig. S.2 in the supplement) confirmed that ozone increases at lower pressure levels and decreases at higher pressure levels, confirming the finding of previous studies. A detailed quantitative analysis is

however out of the scope of the present study.

Figure 9 shows the trends of total ozone (from MERRA-2) and GPH in a wider spatial scale for the months of April and September for which statistically significant changes in total ozone were found. These plots suggest that changes over Rome are part of changes taking place over wider spatial scales. Although deriving trends of the total ozone from MERRA-2 is more uncertain than deriving trends from good quality ground-based measurements (Zhao et al., 2021), the graphs in Fig. 9 still

show clearly that for the considered months (April and September), negative trends in total ozone coincide with positive trends in the GPH (which denotes shift of the tropopause towards higher altitudes) over wide areas. This was not the case for October for which changes in GPH did not coincide spatially with changes in total ozone. Although it is not safe to draw quantitative conclusions from this analysis, it indicates that dynamical changes in the atmosphere could possibly play a significant role in affecting ozone changes.



## 4 Summary and conclusions

In the present study the variability of solar UV irradiance at 307.5 nm and 324 nm, and the 307.5/324 nm ratio was analysed with respect to the variability of total ozone and GPH. Analyses were performed for the sites of Rome, Aosta, and Lampedusa, where long-term and high quality measurements of spectral irradiance and total ozone are available for the period 2006 – 2020. For Rome, analyses were also performed for 1996 – 2020.

Statistically significant anti-correlation was found between the year-to-year variability of GPH and total ozone for all three sites and all seasons of the year, which confirms the findings of studies reporting that variability in troposphere dynamics is strongly correlated with the variability in total ozone (e.g., Ball et al., 2019; Staehelin et al., 2001). The anti-correlation between total ozone and GPH induced a corresponding correlation between GPH and the 307.5 nm irradiance, and between GPH and the 307.5/324 nm ratio, which in most cases was significant. Correlation between GPH and the 307.5 nm irradiance (and GPH and the 307.5/324 nm ratio) was stronger for Aosta relative to Rome and Lampedusa, which can be attributed (at least partially) to the fact that total ozone at higher latitudes generally reaches higher values. Thus, variations in GPH induce larger absolute changes in total ozone and subsequently the UVB irradiance. At Rome, the GPH was also correlated significantly with the 324 nm irradiance (for anomalies in 2006 – 2020 the correlation coefficients for all seasons were ~ 0.4), possibly because changes in GPH were also linked to changes in clouds and aerosols.

The correlation between the GPH at Rome and Aosta was strong (~0.9) and showed that the two sites were frequently affected by the same synoptical systems. The strong correlation between GPH at the two sites corresponded to a strong correlation between total ozone (~0.67), the 307.5 nm irradiance (~0.74), and the 307.5/324 nm ratio (~0.75) at the two sites. Statistically significant, but weak, correlation was also found for the 324 nm irradiance at Rome and Aosta (~0.26). The correlation between GPH at Rome and Lampedusa was again strong and significant (~0.75), while the correlation between GPH at Lampedusa and Aosta was also statistically significant but weaker (~0.48). In both cases the correlation for total ozone was statistically significant, ~0.5 and ~0.4 respectively, but the correlation for the irradiance at 307.5 nm and the 307.5/324 nm ratio was weak, although in some cases it was statistically significant.

The strong correlation between the monthly anomalies of the GPH (and subsequently between the total ozone anomalies) in Aosta, Rome, and Lampedusa was depicted to their long-term trends in 2006 – 2020 which were toward the same direction (i.e, the GPH was increasing/decreasing at all sites) for most months of the year. However, the long-term trends of GPH and total ozone were not statistically significant in most cases. For SZA = 67° the irradiance at 307.5 nm and 324 nm did not change significantly in 2006 – 2020 at none of the three sites. However, the UV irradiance at 307.5 nm and 324 nm increased significantly by ~ 0.7%/year, in Rome at SZA = 45° possibly due to changes in clouds and/or aerosols. At Aosta increases of similar magnitude were found for the 307.5 nm and the 324 nm irradiances (only the latter was statistically significant) at 45° in August, which were possibly mostly driven by changes in clouds and/or aerosols. The increase in UV irradiance in August at Aosta coincided with a statistically significant increase in the GPH, which was not however the case for the changes in UV irradiance in April at Rome. Changes possibly driven by clouds and/or aerosols were found at  SZA = 45° and not at SZA =





67°, which can be explained by the fact that clouds and aerosols, both have a stronger effect on the direct relative to the diffuse component of the solar irradiance. The direct component becomes less significant as the SZA decreases, leading to less

significant trends of the irradiance.

The large, statistically significant increase of total ozone at Aosta in November (during 2006 - 2020), by ~0.8%/year induced a negative trend of ~1%/year in the 307.5/324 nm ratio at 67° SZA, which was not however statistically significant. According to di Sarra et al., (2002) a change of ~1% in total ozone at that SZA should induce a change of 2 – 3% in the irradiance at 307.5 nm. The difference between the observed and the expected change in the irradiance at 307.5 nm can be attributed to the

fact that ozone changed differently at different levels in the atmosphere.

Changes in troposphere dynamics in April and September, which took place on a wide spatial scale during 1996 – 2020, led to decreased total ozone over Rome, which in turn resulted to statistically significant increases in the 307.5/324 nm ratio at 45° SZA. Total ozone also decreased in October, although no statistically significant change of GPH was found, resulting again in a significant increase in the 307.5/324 nm ratio at 45° SZA. In all above cases the changes in total ozone and the

307.5/324 nm ratio were in good agreement with the RAF (Radiation Amplification Factor) reported by (di Sarra et al., 2002). The 307.5/324 nm ratio at 67° SZA decreased significantly in March at the same site and for the same period, although no statistically significant change of total ozone was found. In general, the results for Rome for 1996 – 2020 can be explained by the decreasing ozone in the lower stratosphere and the increasing ozone in the upper stratosphere. The two processes are competitive to each other regarding their effect on UVB irradiance, with a stronger influence on UVB of the lower stratospheric

ozone decrease at the small SZA.

Concluding, the present study shows that total ozone over the Northern part of Italy decreased in the last decades mainly because of changes in the dynamics of the troposphere, the effect of which is dominant over the ozone recovery in the upper stratosphere. The overall decrease of total ozone has a stronger effect on the irradiance at 307.5 nm at smaller SZAs, since the relative contribution of upper stratospheric ozone to the attenuation of UVB irradiance becomes more significant with

increasing SZA. Changes in cloudiness and/or aerosols also may alter the levels of UV irradiance at Aosta and Rome. More robust statistical analyses and radiative transfer modelling are necessary in order to quantify the relative contribution of different factors to the short- and long-term changes of the surface solar UV irradiance in Italy, which however is out of the scope of the present study. In any case, this study shows very clearly that changes in the troposphere can play a key role in the formulation of the levels of total ozone and surface UV radiation. Synoptical circulation patterns and changes were found to

be linked, not only to changes in total ozone, but also in UVB, and in some cases in UVA, irradiance. Since significant change in the dynamics of the troposphere are projected for the future (e.g., IPCC, 2013), thorough investigation is necessary in order to determine their possible impact on the future levels of surface UV radiation.



**Data availability**

The ground based total ozone and spectral UV measurements used in this study can be made available upon request to the PIs

of the stations. All re-analysis and satellite products used in the present study have been downloaded from the Giovanni

platform maintained by National Aeronautics and Space Administration (NASA) (https://giovanni.gsfc.nasa.gov/giovanni/).

**Author contributions**

IF and HD conceptualized the paper. IF led the paper preparation and performed the data analysis. HD is responsible for the

ground-based total ozone and spectral UV measurements in Aosta. AMS is responsible for the ground-based total ozone and

spectral UV measurements in Rome. AdS, DM, and DS are responsible for the ground-based total ozone and spectral UV

measurements in Lampedusa. All authors contributed to the writing of the paper.

**Acknowledgements**

Measurements at Lampeudusa were maintained thanks to support from the Italian Ministry for University and Research

through projects NextData and Marine Hazard, and the Italian Space Agency through the PRISCAV project. Observations also

contribute to the Aerosol, Clouds and Trace Gases Research Infrastructure. Contributions by Francesco Monteleone and

Giandomenico Pace are grate-fully acknowledged.  Part of the analyses used in this paper were produced with the Giovanni

online data system, developed and maintained by the NASA GES DISC.

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





**Figures**

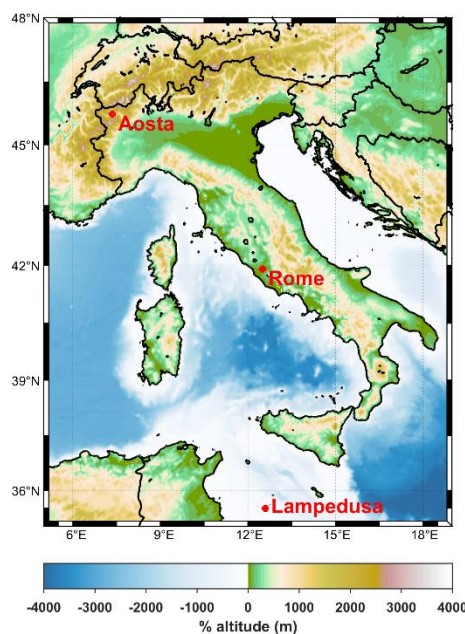

**Figure 1: Topographic map of Italy and the three sites for which measurements are analysed in the study.**





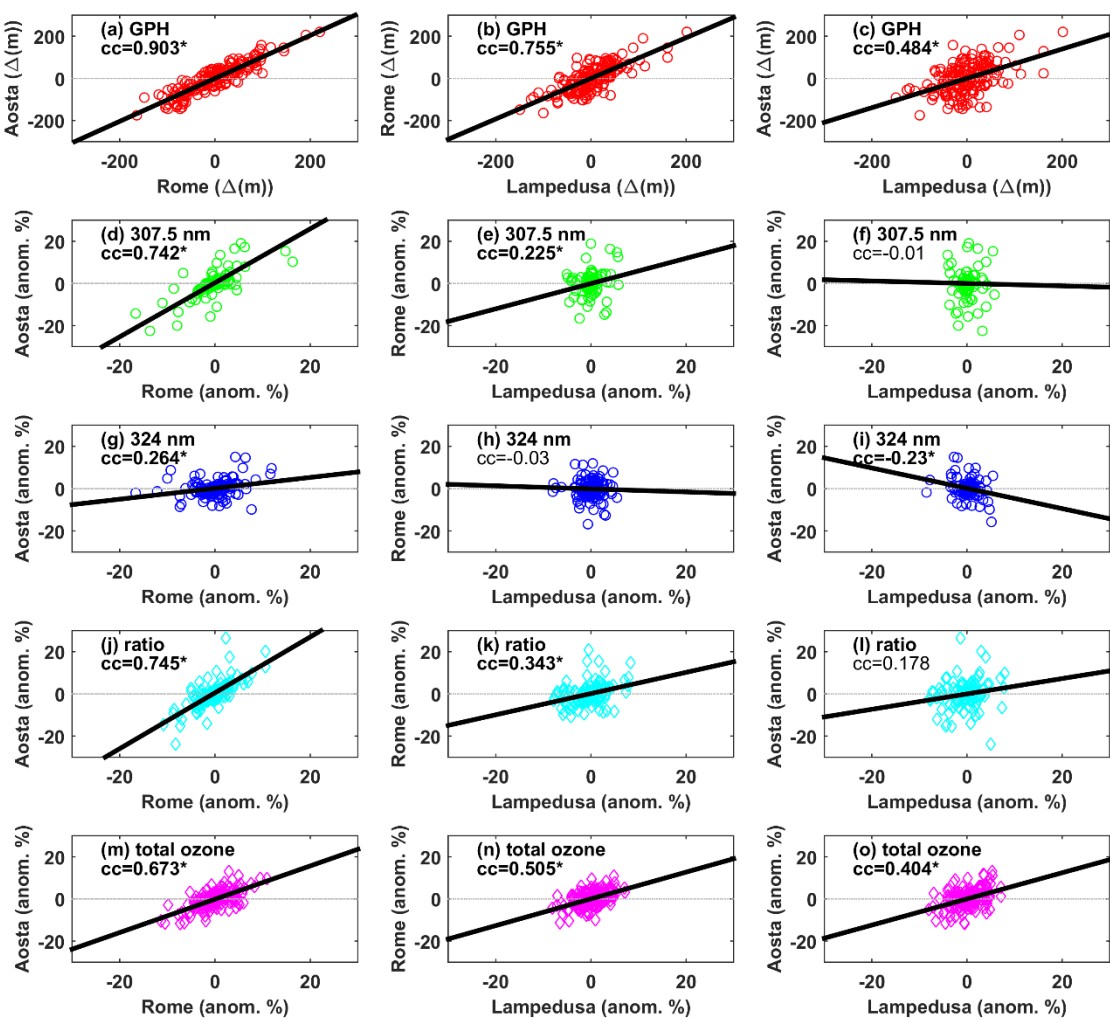


**Figure 2: Scatter plots and correlation coefficients (cc) between the monthly anomalies of GPH (first row), irra-diance at 307.5 nm (second row), irradiance 324 nm (third row), 307.5/324 nm ratio (fourth row), and total ozone (fifth row) in terms of anomalies for the pairs Rome – Aosta (first column), Lampedusa – Rome (second column), and Lampedusa – Aosta (third column). Numbers in bold with asterisk denote statistical significance.**



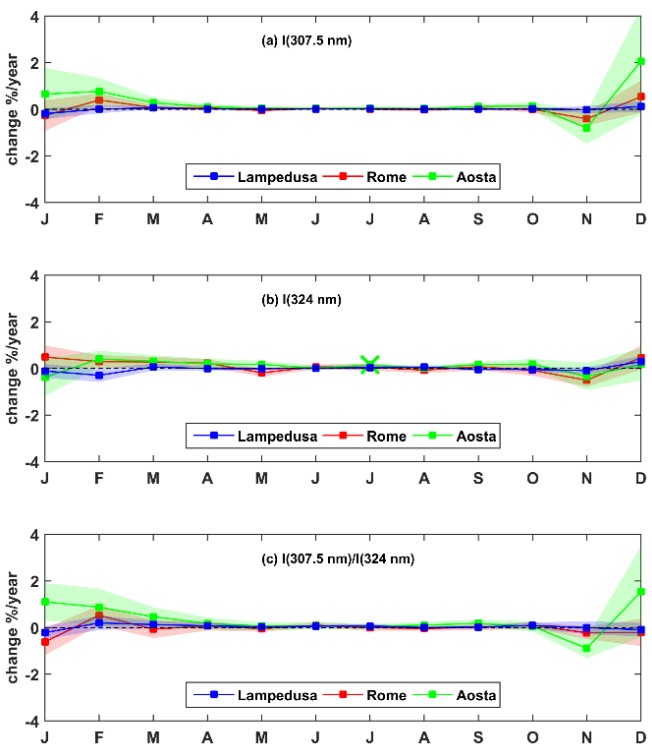


**Figure 3: Average change (%) per year of (a) irradiance at 307.5 nm, (b) irradiance at 324 nm, and (c) the 307.5/324 nm ratio. Results are for SZA=67° for the period September 2006 – August 2020. Statistically significant changes have been marked with x. Shaded areas correspond to the 1-sigma standard deviation.**





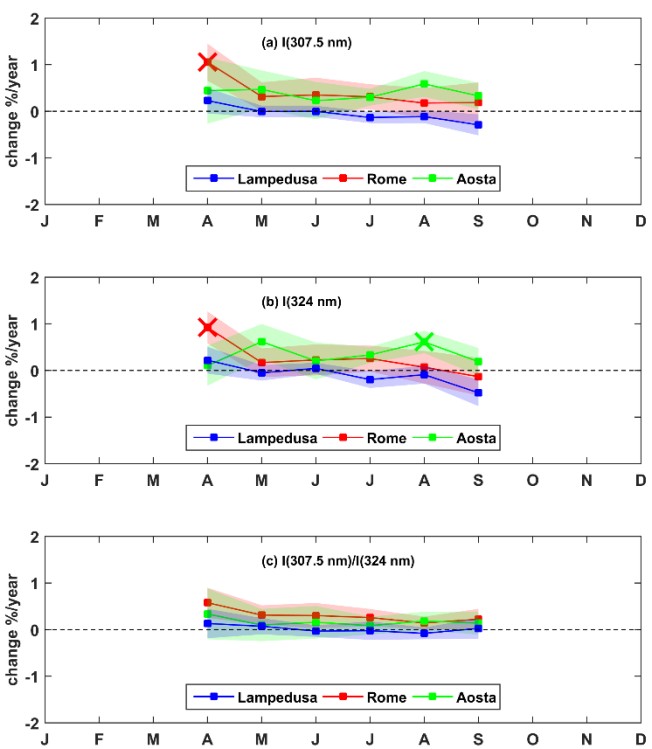

**Figure 4: Average (%) change per year of (a) irradiance at 307.5 nm, (b) irradiance at 324 nm, (c) the ratio between the irradiances at 307.5 nm and 324 nm. Results are for 45° the period 2006 – 2020. Statistically significant changes have been marked with x. Shaded areas correspond to the 1-sigma standard deviation.**





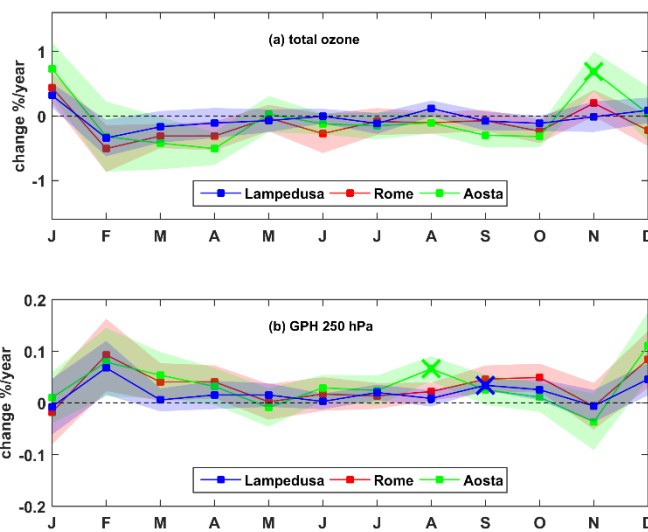

**Figure 5: Average (%) change per year of (a) total ozone, and (b) GPH for the period 2006 – 2020. Statistically significant changes**
**have been marked with x.**

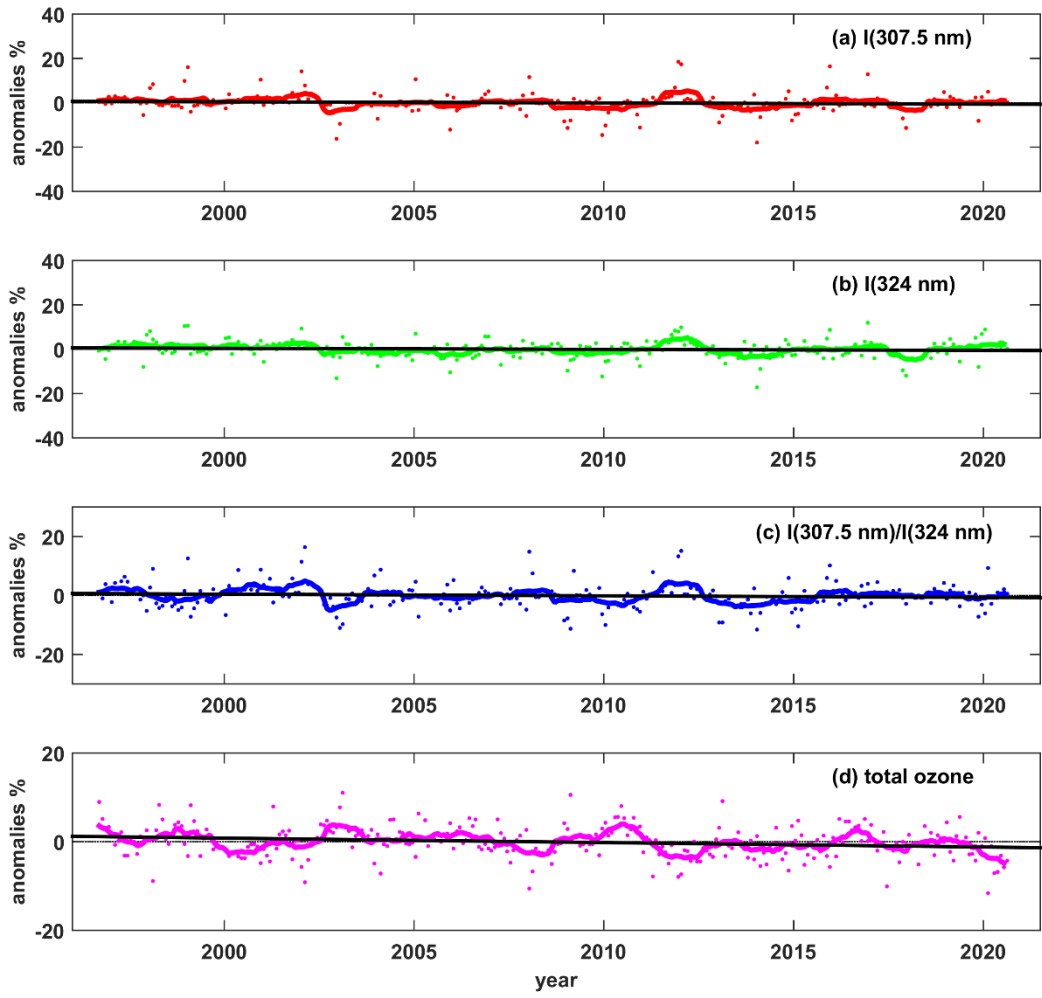

**Figure 6.** Anomalies (% differences relative to the monthly climatological values, represented by dots), 12-month moving averages (thick coloured lines), and trends (black lines) for Rome, for the monthly averages of (a) the irradiance at 307.5 nm, (b) the irradiance at 324 nm, (c) the ratio between the 307.5 nm and 324 nm irradiances, and (d) the total ozone.





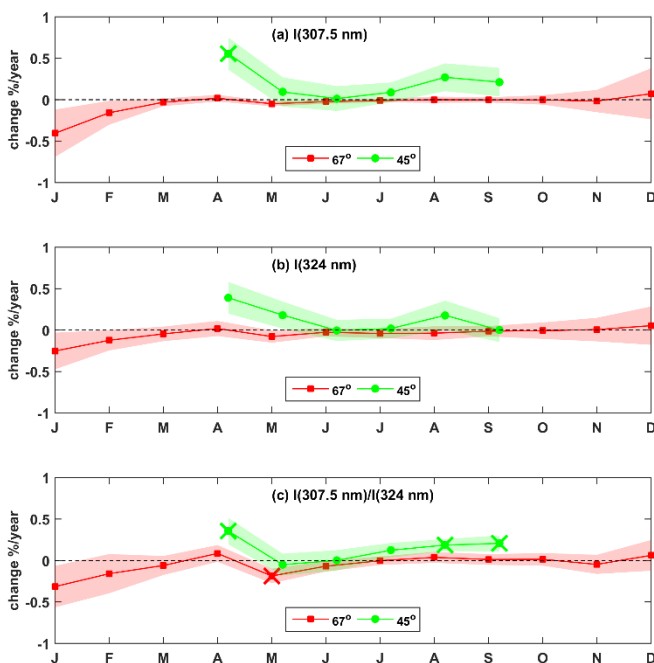

**Figure 7: Average (%) change per year of (a) irradiance at 307.5 nm, (b) irradiance at 324 nm, (c) the ratio between the irradiances at 307.5 nm and 324 nm, for Rome. Trends of the irradiances and the ratio are presented for the 45° and 67.5° SZAs. Statistically significant trends have been marked with x. Shaded areas correspond to the 1-sigma standard deviation.**

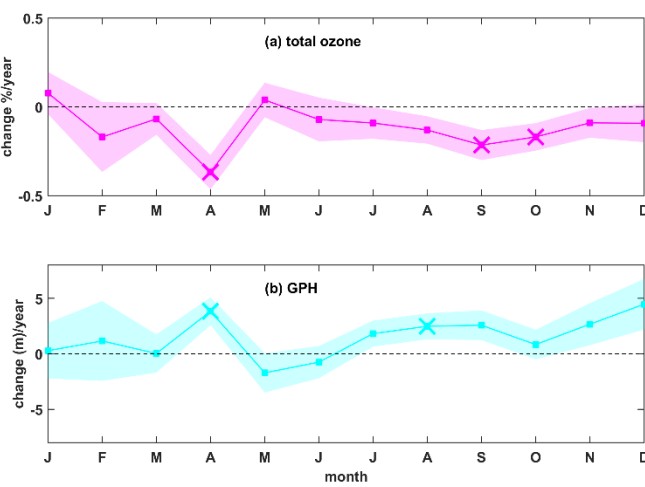


**Figure 8: Average (%) change per year of (a) total ozone, and (b) GPH, for Rome. Statistically significant trends have been marked with x. Shaded areas correspond to the 1-sigma standard deviation.**





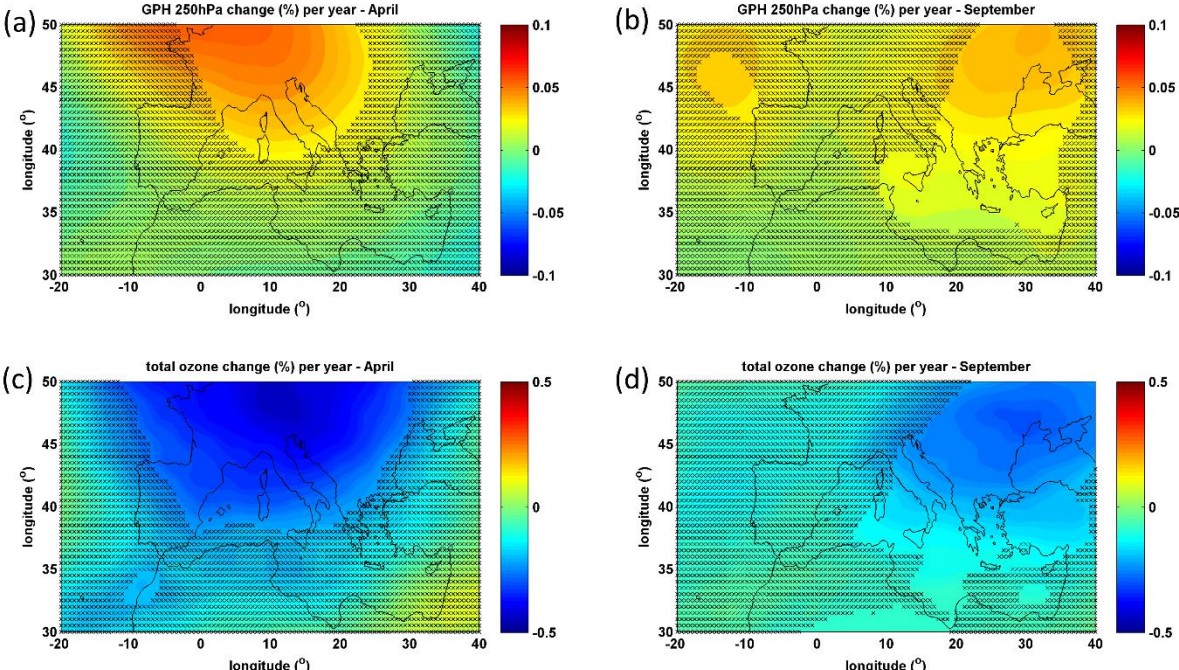

**Figure 9: Changes (%) of GPH at 250 hPa (a,b) and total ozone (c,d) and for April (a,c), and September (b,d). Areas over which changes are not statistically significant are covered by x.**





**Tables**

**Table 1: Correlation coefficients between the anomalies of GPH and other parameters (total ozone, irradiance at 307.5 and 324 nm and 307.5/324 nm ratio) for four different seasons of the year. Values in bold de-note statistically significant correlation or anti-correlation.**

| | Lampedusa | Rome (2006-2020) | Rome (1996-2020) | Aosta |
|---|---|---|---|---|
| | Total ozone | | | |
| Winter | **-0.69** | **-0.69** | **-0.74** | **-0.63** |
| Spring | **-0.64** | **-0.46** | **-0.56** | **-0.68** |
| Summer | **-0.54** | **-0.56** | **-0.49** | **-0.30** |
| Autumn | **-0.51** | **-0.62** | **-0.70** | **-0.70** |
| | 307.5 nm irradiance (45°) | | | |
| Winter | - | - | - | - |
| Spring | **0.51** | **0.59** | **0.57** | **0.60** |
| Summer | **0.41** | **0.55** | **0.46** | **0.44** |
| Autumn | **0.49** | 0.33 | **0.58** | 0.26 |
| | 307.5 nm irradiance (67°) | | | |
| Winter | **0.55** | **0.52** | **0.53** | **0.55** |
| Spring | **0.51** | **0.40** | **0.47** | **0.58** |
| Summer | **0.58** | **0.49** | 0.19 | **0.38** |
| Autumn | **0.52** | **0.64** | **0.48** | **0.64** |
| | 324 nm irradiance (45°) | | | |
| Winter | - | - | - | - |
| Spring | 0.09 | **0.49** | **0.42** | 0.36 |
| Summer | -0.03 | **0.48** | **0.42** | 0.24 |
| Autumn | 0.12 | 0.14 | 0.34 | -0.42 |
| | 324 nm irradiance (67°) | | | |
| Winter | **0.50** | **0.44** | **0.38** | 0.10 |
| Spring | 0.08 | **0.44** | **0.40** | 0.31 |
| Summer | 0.10 | **0.42** | 0.20 | -0.02 |





| | | | | |
|---|---|---|---|---|
| Autumn | 0.07 | **0.42** | **0.33** | **0.41** |
| | 307.5/324 nm ratio (45°) | | | |
| Winter | - | - | - | - |
| Spring | **0.65** | - | **0.52** | **0.69** |
| Summer | **0.61** | **0.52** | **0.39** | **0.54** |
| Autumn | **0.62** | 0.24 | **0.53** | **0.76** |
| | 307.5/324 nm ratio (67°) | | | |
| Winter | **0.54** | **0.44** | **0.62** | **0.67** |
| Spring | **0.44** | 0.19 | **0.39** | **0.66** |
| Summer | **0.59** | **0.50** | 0.17 | **0.55** |
| Autumn | **0.54** | **0.69** | **0.51** | **0.83** |

**Table 2: Trends and statistics for the 307.5 nm irradiance, the 324 nm irradiance, the 307.5/324 nm ratio, and the total ozone for Rome in 1996 - 2020. Significance level is set at 0.05. Statistically significant trends are in bold.**

| | Change %/year | Standard deviation | T-statistic | p-value |
|---|---|---|---|---|
| 307.5 nm | -0.04 | 0.04 | -1.30 | 0.20 |
| 324 nm | -0.04 | 0.04 | -1.30 | 0.20 |
| 307.5/324 nm | -0.06 | 0.04 | -1.42 | 0.16 |
| TOC | **-0.10** | 0.03 | -3.21 | <0.01 |