# Peer review of "Variability and trends in surface solar spectral ultraviolet irradiance in Italy: on the influence of geopotential height and lower stratospheric ozone"

_Atmospheric Chemistry and Physics, 2021_

## Author Comment (AC1)

Reply to anonymous reviewer#1

We gratefully acknowledge anonymous reviewer#1 for his/her comments, that helped us improve the manuscript substantially. We tried to address all reviewer's comments. In the following, analytical replies are provided to each of the reviewer's comments. Reviewer's comments are written in bold font. Line numbers, when provided refer to the version with track changes.

**General comments**

**The manuscript by Fountoulakis et al. correlates total ozone columns (TOC) and surface UV radiation at 307.5 and 324 nm with geopotential height at 250 hPa (GPH). While the anti-correlation between tropopause height and TOC has long been known, using GPH instead of tropopause height is a novel idea. Using GPH, the authors then explore the effect of atmospheric pressure patterns on short- and long-term variations in ozone and UV radiation. The results are interesting and worth publishing in ACP. In the second part of the paper, the authors then analyze UV measurements at three Italian stations for long-term trends and use TOC and GPH data to interpret these trends.**

**Most of the analyses are sound, with the exception of two issues: the effect of the atmospheric ozone profile on UV radiation and the effect of changes in aerosol and clouds on trends, as described in "Mayor Comments" below. The presentation is generally clear, although the Summary at the end is too long; too much detail distracts from the main messages. I recommend publication of the manuscript, provided that my Mayor and Minor comments below are addressed appropriately.**

Reply

We tried to shorten the Summary at the end of the manuscript as suggested by the reviewer. After further analysis we found that the reviewer was right about the two issues described above, and we tried to correct the manuscript properly. More details are provided in the replies to the reviewer's specific comments.

**Major Comments:**

**The authors mention multiple times that the vertical redistribution of ozone in the atmosphere has an important effect on the global irradiance at 307.5 nm and that changes of this distribution over time can explain some trends in their UV measurements. My model calculations (see below) suggests that this effect is not important for the solar zenith angles (SZAs) of 45° and 67° considered in this study. The authors should perform their own model calculations and adjust their assertions accordingly.**

Reply

We performed model simulations as recommended by the reviewer, and indeed, the effect of changes in total ozone is larger at the SZA of 67° with respect to 45°, independently from the vertical distribution of changes in total ozone (at least for changes up to 10% with respect to the average levels of total ozone in Italy). However, as also suggested by the reviewer, we repeated the analysis of the long-term changes of the spectral UV irradiance, more carefully this time, and our new findings are in agreement with the modelling simulations.

**Along the same lines, the authors imply that clouds and aerosols have a larger effect on surface UV radiation at a SZA of 45° compared to 67°. For example, they report that there are no significant trends in UV radiation at 67° at Rome while significant trends were calculated for 45° (their Figure 7). They attribute this difference to cloud and aerosol effects. My model calculations**

(also below) confirm that aerosol effects are larger at 67° than 45°. It is therefore unlikely that there is no trend at 67° but a signficant trend at 45° because of cloud or aerosol effects. Instead, I suspect that there is a problem in the UV radiation data at 45°. However, the data presented do not allow me to confirm this suspicion.

Reply

Under certain conditions attenuation by clouds and aerosols can be larger at lower SZAs (see Figures S1 and S2 in the supplement), especially in UVB wavelengths. However, this was not the case in our study. As correctly suggested by the reviewer there was a problem with the derived UV trends (they were significantly affected by gaps in the series), but at SZA=67°. In order to solve this issue, we tried to improve the filtering of the data used in the analysis. Furthermore, for Rome and Lampedusa we performed the analysis for the SZA of 65° instead of 67°, which gave more reliable results (see lines 196-206 of the manuscript). For Aosta data availability was better at 67°, thus analysis was performed for the particular SZA. Despite the small difference in the SZA for which the data were analyzed results for the three stations are comparable to each other as explained in the manuscript.

**Minor Comments:**

**L54: Regarding "avoided extremely high levels of solar UVB radiation". Perhaps give a number here.**

Reply

Relative information has been added to the manuscript.

**L56: The recent _Nature_ paper by Young et al. (https://doi.org/10.1038/s41586-021-03737-3) could also be cited here.**

Reply

Done

**L70: You could also cite: Ohring & Muench 1960: (https://doi.org/10.1175/1520-0469(1960)017<0195:RBOAMP>2.0.CO;2).**

Reply

Done

**L72: Please provide a reference that the tropopause rises with warming of the troposphere.**

Reply

Done

**L152: regarding "standard uncertainty for spectral UV measurements … of the order of 5%" So that would be 10% at the 2-sigma level?**

Reply

Yes, this is correct. Throughout the manuscript we refer to 1-sigma (standard) uncertainties, which in our opinion is clear. Thus, we do not believe that further discussion is necessary in the manuscript relative to the 2-sigma uncertainty level.

**L156: "smaller than 2.5%" is only half as large as the 5% quoted above. Why?**

Reply

The uncertainty of 5% discussed above refers to Brewer spectrophotometers. The 2.5% uncertainty at this point refers to the Bentham spectroradiometer operating at Aosta. We think that this is already clear in the manuscript.

**L168: Regarding "available for at least 15 days.": That's only half the days in a month. Significant error in monthly averages could occur if the missing days are biased towards either the start or end of a month. How was this problem addressed? Could this have caused spurious trends?**

Reply

The reviewer is right at this point. The threshold of 15 days was used as a compromise between representativeness and availability of data to perform a trend analysis. To make sure that there is no bias issue due to the distribution of gaps we changed the data filtering method for the data used in the revised version of the manuscript. Again, we only consider months for which measurements are available for at least 15 days, with the additional criterion that at least 5 days are available for each of the following sub-periods in the month: day 1 – day 10, day 11 – day 20, day 21 – end of month (see line 208).

**L225: The study focuses on anomalies in GPH at 250 hPa rather than tropopause altitude, the parameter frequently in other studies (see line 70). Figure S1 in the Supplement shows that there is a good correlation between the two parameters. Please describe the advantage of using GHP instead of tropopause height considering that the tropopause separates tropospheric and stratospheric ozone and therefore might be the more important parameter.**

Reply

In this study we studied the link between the GPH at 250 hPa and total ozone in order to investigate and highlight the relationship between synoptical tropospheric conditions (for which the 250 hPa GPH is more representative relative to the altitude of the tropopause) and total ozone. The same information has been added to the manuscript (lines 224 - 230).

**L253: Attributing correlation coefficients between GPH and the other parameters shown in Table 1 to "dynamical stratospheric processes" and "tropospheric processes" is a bit of a stretch. It would be more appropriate to say that a higher GPH emphasizes processes in the troposphere while a lower GPH emphasizes those in the stratosphere.**

Reply

We deleted this phrase in the revised version of the manuscript.

**L263 - 285: Please structure better: First describe changes at SZA=67° based on Fig 3, then do the same for SZA=45°, based on Fig 4. Lastly, highlight the differences between trends at the two SZA.**

Reply

We tried to improve the structure of this section.

**L263-270: The text does not fit Figure 3. The whole paragraph is questionable, and while it identifies Figure 3, it should be mentioned that this description refers to the analysis of data at SZA=67°. Specifically, "307.5" in line 263 should be "324"; "324" in line 264 should be "307.5". Regarding "The overall increase of the 307.5 irradiance for 2006 – 2020": Do you refer to an increase averaged over all months? If so, how was the annual anomaly calculated considering the large difference in winter and summer UV radiation?**

Reply

Obviously, the manuscript was confusing at this point. In this section trends have been only studied and reported separately for each month of the year, and not as average over all months. We re-wrote this part of the manuscript clearer for the results of the updated analysis.

**Lines 271 - 285: The text should be better structured. First, it should be said that this analysis is now for SZA=45° and that data from Aosta, Lampedusa, and Rome are discussed sequentially. For example: "We now discuss changes observed at a SZA of 45°. At Aosta, the irradiance at 324 nm increased by 0.6%/year in August..."**

Reply

The 3.2.1 section was structured in a different way following the recommendation of the reviewer.

**L275 - 277 and L351 - 353: As already noted above, I find it hard to believe that there were no trends at 67° but significant trends at 45° due to cloud and aerosol effect. Clouds and aerosols typically have a larger effect at 67° compared to 45°. To quantify this, I modeled spectra of global irradiance for 45° and 67°, either without aerosols or by assuming an aerosol layer. I parameterized the aerosol optical depth with Angström's formula, setting alpha=1 and beta=0.25. This is a rather dense aerosol layer. At SZA=45°, global irradiance with aerosols was suppressed relative to the no-aerosols case by 17.5% at 307.5 nm and 16.3% at 324 nm. At 67°, global irradiance was lower by about 20% at both wavelengths, confirming that the effect of aerosols increases with SZA. Hence, I do not believe that clouds or aerosol are responsible for the different trends at 45° and 67°. Instead, I suspect some problems in the data, e.g., due to gaps. This issue should be explored further by the authors with their own model calculations.**

Reply

The reviewer has a point here. This issue has been addressed properly as discussed earlier (see replies to the reviewer's general comments). We are thankful to the reviewer, who helped us solve this significant issue and improve the manuscript.

**L284: Considering that the changes for April and May are so different, can it be ruled that problems in the data, such as data gaps affecting the calculation of monthly averages, led to the high value in April?**

Reply

We checked and did not find data gaps that could be responsible for the differences between April and May. The difference between April and May can be safely attributed to the corresponding difference in the trends of total ozone. Analysis of total ozone series using Brewer and MERRA-2 data for Rome (analysis of the MERRA-2 ozone is not discussed in the manuscript) yielded similar results: no trend in May, and significant decrease in April.

**L288 and L294: Why suddenly "SZA of 67.5°"? Up to now the SZA was 67°.**

Reply

It was a typo and has been corrected.

**L300: Again, I find it hard to believe that the trends for a SZA at 45° and 67° (or 67.5° ?!?) are so different, in particular for April, and to a lesser extent for August and September. The fact that the large trend for April is also present at 324 nm suggests that the trend in ozone (Fig 8a) is not the driving factor.**

Reply

After improving and updating the analysis we found similar results for the two SZAs, which were indeed attributed to aerosols and clouds and not ozone.

**L310: Regarding: "As the SZA increases the role of ozone at the middle and upper atmosphere becomes more important regarding the attenuation of the UV-B irradiance relative to ozone at the lower stratosphere." I presume that you refer to the Umkehr effect here. However, that effect is only significant for SZA > 80°. To confirm that the ozone profile has only a minor affect on UV irradiances at 307.5 and 324 nm for SZA of 45° and 67°, I ran model calculations (LibRadtran/UVSPEC) using either the standard mid-latitude profile (afglms.dat) or a modified profile where I increased the ozone concentration by 5% in the upper troposphere and lower stratosphere (between 13 and 20 km). I then scaled the original afglms profile and this modified profile to a TOC of 315 DU. This scaling effectively increased the ozone of the modified profile by 3.8% between 13 and 20 km and lowered it by 1.1% at all other altitudes. The global spectral irradiance at 307.5 nm calculated with the modified profile for SZA=45° was 0.06% larger than the irradiance calculated with the standard profile. The difference for 67° was 0.09%. At 324 nm, the difference was basically zero. These calculations show that the effect of the profile at these SZAs is negligible. So the sentence in line 310 should be removed.**

Reply

The reviewer is right. We performed similar analysis and we got similar results as the reviewer. Again, we thank the reviewer for his/her efforts. The manuscript has been corrected.

**L326-375: The summary is too long. Please shorten and emphasize the essential numbers and messages rather than repeating the Results section.**

Reply

The summary and conclusions section has been shortened significantly as suggested by the reviewer

**L355: The sentence "The increase … at Rome" is one example of a sentence that does not add much and could be deleted to make Section 4 more focused.**

Reply

It has been deleted

**L358: As mentioned above, my model calculations do not support the assertion that the difference at 45° and 67° is due to clouds and aerosols. If the authors feel otherwise, they should support their assertion with their own calculations.**

Reply

We have replied to this comment earlier

**L359: "SZA decreases" > "SZA increases". ("SZA decreases" means that the Sun is closer to the zenith and the contribution of the direct irradiance becomes larger, not "less significant".)**

Reply

Corrected

**L364: My model calculations above strongly suggest that the following sentence is either incorrect or greatly overstates the effect: "The difference between the observed and the expected change in the irradiance at 307.5 nm can be attributed to the fact that ozone changed differently at different levels in the atmosphere."**

Reply

Appropriate corrections have been applied to the manuscript.

**L372: The same can be said about the sentence "… can be explained by the decreasing ozone in the lower stratosphere and the increasing ozone in the upper stratosphere."**

Reply

This sentence has been removed

**L379: Again, at 67° the effect of the profile is negligible and the sentence "of upper stratospheric ozone to the attenuation of UVB irradiance becomes more significant with increasing SZA" should be removed.**

Reply

The sentence has been removed.

**L380: I agree that. "More robust statistical analyses and radiative transfer modelling are necessary in order to quantify the relative contribution of different factors to the short- and long-term changes of the surface solar UV irradiance in Italy" but I disagree that these calculations are "out of the scope of the present study." The authors assert multiple times that the vertical redistribution of ozone has an import effect on global UV irradiance for SZA of 45° and 67°, contrary to my calculations. They should make their own calculations to look into this issue in more detail than I did, and perhaps add an additional figure to the manuscript summarizing their results.**

Reply

More robust statistical analysis has been performed and the manuscript has been updated according to the findings of this analysis,

**Technical suggestions / grammar / typos:**

**L13: Delete "In this study" (It is obvious that the abstract refers to this study.)**

Reply

Done

**L14: over Italy > across Italy (Otherwise readers might think that UV was measured in the air above ground.)**

Reply

Done

**L14: "located at quite different" > "characterized by quite different"**

Reply

Done

**L16: "307.5 nm, 324 nm" > "307.5 nm and 324 nm"**

Reply

Done

**L18: "geopotential height at 250 hPa (GPH)." > "geopotential height (GHP) at 250 hPa."**

Reply

Done

**L28: "It was also showed that" > "It was also shown that" or "We also showed that"**

Reply

Done

**L31: "period which" > "period, which" ; "aerosols were" > "aerosols, were"**

Reply

Done

**L42: "is also absorbed" > "is absorbed"**

Reply

Done

**L43: "ozone dominates on scattering" > "ozone has greater importance than scattering**

Reply

We changed the phrase "dominates on" with "is more significant than"

**L46:** "leading to reduced" > " but also reduced" (otherwise it sounds as if low- and mid-latitude ozone changes are only the result of high-latitude processes)

Reply

Done

**L52:** "resulted on decreasing" > "resulted in decreasing"

Reply

Done

**L89:** "parameters which affect significantly" > "parameters that significantly affect"

Reply

Done

**L96:** "different latitudes and environmental conditions" > "and affected by differing environmental" (a site cannot be "located at an environmental condition")

Reply

Done

**L97:** "extent at which" > "extent to which"

Reply

Done

**L118:** "referred as IOS standard." > "referred to as IOS standard."

Reply

Done

**L188:** "GPH at 250 hPa" > "GPH at 250 hPa and at 850 hPa", and delete "and ...850 hPa."

Reply

Done

**L243:** "For 45°" > "For a SZA of 45° "

Reply

Done

**L279:** "none" > "any"

Reply

Done

**L286: "3.2.1 Long-term variability in the period 1996 – 2020" > "3.2.1 Long-term variability at Rome for the period 1996 - 2020"**

Reply

Done

**L348: "of the GPH" > "in the GPH"**

Reply

Done

**L349: I don't understand "was depicted to"**

Reply

We re-wrote the sentence in a clearer way.

**Supplement, line 3: "Figure A1" > "Figure S1"**

Reply

Corrected

**Supplement, line 4: GHP at 250 hPa and 850 hPa are correlated, not "anti-correlated"**

Reply

Corrected

**Supplement, Caption Figure S1: "tropo-pause" > "tropopause"; delete extra space after 850 hPa**

Reply

Corrected

**Variability and trends of the surface solar spectral ultraviolet irradiance in Italy: on the combined influence of dynamics and lower and upper stratospheric ozone trends**

Ilias Fountoulakis[1,a], Henri Diémoz[1], Anna Maria Siani[2], Alcide di Sarra[3], Daniela Meloni[3], Damiano M. Sferlazzo[4]

[1]Aosta Valley Regional Environmental Protection Agency (ARPA) of the Aosta Valley, 11020 Saint-Christophe, Italy
[2]Physics Department, Sapienza Università di Roma, 00185 Rome, Italy
[3]ENEA, Laboratory for Earth Observations and Analyses, 00123, S. Maria di Galeria, Rome, Italy
[4]ENEA, Laboratory for Earth Observations and Analyses, 92010, Lampedusa, Italy
[a]Now at: Institute for Astronomy, Astrophysics, Space Applications and Remote Sensing, National Observatory of Athens (IAASARS/NOA), 15236 Athens, Greece

*Correspondence to*: Henri Diémoz (h.diemoz@arpa.vda.it)

**Abstract.** The short- and long-term variability of the surface spectral solar ultraviolet (UV) irradiance is investigated across Italy using high quality ground-based measurements from three sites characterized by quite different environmental conditions and covering the full latitudinal extent of the Italian territory: Aosta (45.7° N, 7.4° E, 570 m a.s.l.), Rome (41.9° N, 12.5° E, 75 m a.s.l.), and Lampedusa (35.5° N, 12.6° E, 50 m a.s.l.). The variability of the irradiances at 307.5 nm and 324 nm, and their ratio are investigated with respect to the corresponding variability in total ozone and the geopotential height (GPH) at 250 hPa.  The study was performed for two periods: 2006 – 2020 for all stations, and 1996 – 2020 only for Rome. A statistically significant correlation between the GPH and total ozone monthly anomalies was found for all stations and all seasons of the year. A corresponding statistically significant correlation was also found in most cases between the GPH and the 307.5 nm irradiance monthly anomalies. The correlation among the GPH monthly anomalies at the three sites was statistically significant, possibly explaining the strong and significant correlation among the corresponding total ozone monthly anomalies. A statistically significant  total ozone decrease, of ~0.1%/year was found for Rome over the period 1996 – 2020, which however at SZA = 65° did not lead to increasing trends in the annual levels of the irradiance at 307.5 nm (neither increasing trends in the corresponding ratio between the 307.5 nm and the 324 nm irradiances). Further analyses revealed positive trends in the ratio and the 307.5 nm irradiance in particular months which followed negative trends in total ozone. The  total ozone decrease over Rome was mainly attributed to decreasing  lower  stratospheric ozone. Furthermore, it was shown that over all stations long-term changes in total ozone follow changes in GPH, which is an additional indication that negative trends in total ozone were mainly driven by changes in

lower stratospheric ozone. An anti-correlation between the GPH long-term changes and total ozone was also evident for all stations in the period 2006 – 2020. For specific months positive trends in UV irradiance for this latter period, which

35    were mostly driven by changes in clouds and/or aerosols rather than total ozone, were found for each of the three sites. This study clearly points out the 
[revised manuscript text omitted]
 for summer months, of 0.05%/year and 0.2%/year for 307.5 nm and 324 nm respectively, possibly driven by changes in aerosols.  At Aosta increases of ~0.7%/year were found for the 307.5 nm and the 324 nm irradiances (only the latter was statistically significant) at 45° in

440 August, which were possibly  driven mostly by changes in clouds and/or aerosols. Changes in troposphere dynamics in April and September, which took place on a wide spatial scale during 1996 – 2020, led to decreased total ozone over Rome, which in turn resulted to statistically significant increases in the 307.5/324 nm ratio at 45° and 65° SZA. The ratio also increased in August following the (not statistically significant) decrease in total ozone. In October the ratio increased, though not significantly, following the significant 
[revised manuscript text omitted]

**Supplement**

**Attenuation by clouds**

In order to investigate the spectral effect of different cloud types on spectral solar UV irradiance in the range 305 – 324 nm we performed simulations with the model uvspec of the libRadtran package (Mayer and Kylling, 2005). Simulations were performed for total ozone equal to 330 DU, AOD=0.1, and for two types of clouds: typical high altitude cirrus clouds (base altitude=10 km, width=1km, effective radius=20 μ m) and typical low altitude water clouds (base altitude=2 km, width=2 km, effective radius=10 μm). The ratio between the spectral irradiance simulated for cloudless and cloudy conditions is presented in Figures S1 and S2.

[Figure]

**Figure S1: Ratio between the surface solar spectral irradiance at 305 – 325 nm reaching the Earth surface with and without low altitude clouds.**

[Figure]

**Figure S2: Ratio between the surface solar spectral irradiance at 305 – 325 nm reaching the Earth surface with and without high altitude clouds.**

A Cloud Optical Depth (COD) of 1 was considered for the cirrus cloud (Figure S1), while a COD of 6 was considered for the low altitude cloud (Figure S2). For the low altitude clouds, the spectral shape of the attenuation is similar for the SZAs of 45° and 67° and the dependence from wavelength is in both cases small. For the high altitude clouds the attenuation decreases with wavelength at 45° and increases with wavelength at 67°. Attenuation of the irradiance at wavelengths larger than 320 nm is similar at both SZAs. At wavelengths that are affected stronger by ozone attenuation by high altitude clouds is significantly stronger at 45°. Performing the same analysis for different CODs ranging from 1 to 30 for the two cloud types resulted to similar conclusions. From the above discussion it is evident that changes in the occurrence of cirrus would result to more pronounced effects on the levels of UVB irradiance at smaller SZAs.

**Correlation between GPH at 250 hPa, GPH at 850 hPa and tropopause altitude**

As shown in Figure S3, there is a strong, statistically significant correlation between the GPH at 250 hPa and the tropopause altitude over Aosta, Rome and Lampedusa. As also shown in Figure S1 (panels d- f) GPH at 250 hPa is strongly correlated with GPH at 850 hPa.

[Figure]

**Figure S3: Correlation between the absolute anomalies (in m) between GPH at 250 hPa and tropopause altitude for (a) Aosta, (b) Rome, and (c) Lampedusa, and correlation between the GPH at 250 hPa and 850 hPa for (d) Aosta, (e) Rome, and (f) Lampedusa. The correlation coefficients (cc) are shown at the lower right side of each graph. Values in bold marked with an asterisk denote statistically significant correlation.**

**Ozone trends at different pressure levels**

As shown in Figure S4, ozone over Rome increases significantly at higher stratospheric levels and decreases significantly at lower stratospheric levels.

[Figure]

**Figure S4: Variability and long-term trends (in %) of the ozone mixing ratio at different atmospheric pressure levels over Rome. Average change per year is shown at the lower right of each graph. Values in bold marked with an asterisk denote statistically significant changes.**

**References**

Mayer, B. and Kylling, A.: The libRadtran software package for radiative transfer calculations-description and examples of use, Atmos. Chem. Phys., 5, 1855–1877, 2005.

---

## Author Comment (AC2)

Reply to anonymous reviewer#2

We acknowledge anonymous reviewer#2 for his/her very useful comments, that helped us improve the manuscript. We tried to address all reviewer's comments. In the following, analytical replies are provided to each of the reviewer's comments. Reviewer's comments are written in bold font. Line numbers, when provided refer to the version with track changes.

**General comments:**

**The manuscript studies if atmospheric dynamics plays a role in total ozone and UV irradiance anomalies by analysing correlation between geopotential height (GPH) and total ozone and UV irradiance in Italy. Correlations between three different sites are also studied as well as trends in analysed parameters. The subject is interesting and important regarding possible changes in the future in the atmospheric dynamics due to the climate change. The manuscript is well written and easy to follow, except the Summary and Conclusions which should be changed to be more reader friendly. I enjoyed especially the Introduction which covers in a nice way the background of this study. However, I think the manuscript lack of explanation related to the effect of the dynamic: Why is there such a strong anticorrelation between tropopause height and total ozone? Is the stratosphere, which includes the "ozone layer", somehow smaller when the tropopause is high-> less ozone, or what is the reason? I am not totally convinced about the conclusion that the influence of lower and upper stratospheric ozone trends is seen in the variability of surface UV irradiance. The manuscript would benefit of radiative transfer calculations to confirm the conclusion. The same applies to the explanations for differences in trends between SZA 45deg and 67deg.**

**Reply**

We reduced the size of the "summary and conclusions" section and tried to focus on the main findings of the study. Radiative transfer simulations have been performed which helped us find problems in the analysis of the UV data. In the new version of the manuscript analysis of the UV irradiance has been updated, leading to more reliable results.

Relative to the question: "Why is there such a strong anticorrelation between tropopause height and total ozone?". This is not a simple question to answer and certainly out of the scope of the present study. We tried to add some more relative discussion and references in the introduction (see lines 81 – 91) wherein possible answers and relative discussion can be found.

**It is not clear how the UV time series of Rome have been homogenized. Was the UV scale adjusted to that provided by the QASUME and the IOS instruments during the comparisons? And how was the UV calibration performed before 2003?**

Reply

During the whole period for which measurements are available at Rome, regular stability checks with 50 Watt lamps, and calibrations with 1000 Watt lamps (traceable to NIST) are performed. Relative information has been added to the document (lines 148 - 150). All inter-comparisons with UV reference instruments since 2003 have resulted to differences within the combined uncertainties of the two instruments, and thus no adjustment has been applied.

**Based on the results, would it be possible to give the range (in %) in which total ozone and UV can vary due to changes in dynamics?**

Reply

Useful information for the quantitative estimation of how GPH affects total ozone and UV irradiance has been added in Table 1. Then, this information was used in order to give an estimate of the point at which the trends in total ozone and UV can be explained by changes in GPH (see lines 383 - 385)

**Specific comments:**

**l. 181: Please explain why you chose 250 hPa and 850 hPa, and not some other GPH.**

Reply

Explanation was added.

**l. 187-191: This doesn't belong to Data and Methodology. Please move to an appropriate place.**

Reply

As suggested by the reviewer this part of the manuscript was moved to the introduction.

**l. 359: SZA decreases -> SZA increases ?**

Reply

Corrected

**Fig.2. In some plots you can not really use the linear correlation analysis. E.g., (h), (i), (f).**

Reply

The reviewer is right here. However, as we state in the manuscript we used Pearson method to study the correlation because "For all cases that a strong correlation was found, it was optimally described by a linear fit". For the cases listed by the reviewer the correlation was weak independently of the used method. Thus, we believe that it is meaningless to show the results of the analysis using a different method (e.g., Spearman correlation).

**Supplement: I don't understand the sentence: " As also shown in Figure A1 (panels d-f) GPH at 250 hPa is strongly anti-correlated with GPH at 850 hPa.". To me, the Figure shows positive correlation, not anti-correlation.**

Reply

Corrected.

**Technical comments:**

**I didn't find the Appendix. I found the Supplement yes, but not the Appendix.**

Reply
It was a typo which has been corrected in the revised version. There is no Appendix, only supplement.

**Variability and trends of the surface solar spectral ultraviolet irradiance in Italy: on the combined influence of dynamics and lower and upper stratospheric ozone trends**

Ilias Fountoulakis[1,a], Henri Diémoz[1], Anna Maria Siani[2], Alcide di Sarra[3], Daniela Meloni[3], Damiano M. Sferlazzo[4]

[1]Aosta Valley Regional Environmental Protection Agency (ARPA) of the Aosta Valley, 11020 Saint-Christophe, Italy
[2]Physics Department, Sapienza Università di Roma, 00185 Rome, Italy
[3]ENEA, Laboratory for Earth Observations and Analyses, 00123, S. Maria di Galeria, Rome, Italy
[4]ENEA, Laboratory for Earth Observations and Analyses, 92010, Lampedusa, Italy
[a]Now at: Institute for Astronomy, Astrophysics, Space Applications and Remote Sensing, National Observatory of Athens (IAASARS/NOA), 15236 Athens, Greece

*Correspondence to*: Henri Diémoz (h.diemoz@arpa.vda.it)

**Abstract.** The short- and long-term variability of the surface spectral solar ultraviolet (UV) irradiance is investigated across Italy using high quality ground--based measurements from three sites characterized by quite different environmental conditions and covering the full latitudinal extent of the Italian territory: Aosta (45.7° N, 7.4° E, 570 m a.s.l.), Rome (41.9° N, 12.5° E, 75 m a.s.l.), and Lampedusa (35.5° N, 12.6° E, 50 m a.s.l.). The variability of the irradiances at 307.5 nm and 324 nm, and their ratio are investigated with respect to the corresponding variability in total ozone and the geopotential height (GPH) at 250 hPa. The study was performed for two periods: 2006 – 2020 for all stations, and 1996 – 2020 only for Rome. A statistically significant correlation between the GPH and total ozone monthly anomalies was found for all stations and all seasons of the year. A corresponding statistically significant correlation was also found in most cases between the GPH and the 307.5 nm irradiance monthly anomalies. The correlation among the GPH monthly anomalies at the three sites was statistically significant, possibly explaining the strong and significant correlation among the corresponding total ozone monthly anomalies. A statistically significant  total ozone decrease, of ~0.1%/year was found for Rome over the period 1996 – 2020, which however at SZA = 65° did not lead to increasing trends in the annual levels of the irradiance at 307.5 nm (neither increasing trends in the corresponding ratio between the 307.5 nm and the 324 nm irradiances). Further analyses revealed positive trends in the ratio and the 307.5 nm irradiance in particular months which followed negative trends in total ozone. The  total ozone decrease over Rome was mainly attributed to decreasing  lower  stratospheric ozone. Furthermore, it was shown that over all stations 
[revised manuscript text omitted]
 for summer months, of 0.05%/year and 0.2%/year for 307.5 nm and 324 nm respectively, possibly driven by changes in aerosols.  At Aosta increases of ~0.7%/year  were found for the 307.5 nm and the 324 nm irradiances (only the latter was statistically significant) at 45° in August, which were possibly  driven mostly by changes in clouds and/or aerosols. Changes in troposphere dynamics in April and September, which took place on a wide spatial scale during 1996 – 2020, led to decreased total ozone over Rome, which in turn resulted to statistically significant increases in the 307.5/324 nm ratio at 45° and 65° SZA. The ratio also increased in August following the (not statistically significant) decrease in total ozone. In October the ratio increased, though not significantly, following the significant 
[revised manuscript text omitted]

**Supplement**

**Attenuation by clouds**

In order to investigate the spectral effect of different cloud types on spectral solar UV irradiance in the range 305 – 324 nm we performed simulations with the model uvspec of the libRadtran package (Mayer and Kylling, 2005). Simulations were performed for total ozone equal to 330 DU, AOD=0.1, and for two types of clouds: typical high altitude cirrus clouds (base altitude=10 km, width=1km, effective radius=20 µ m) and typical low altitude water clouds (base altitude=2 km, width=2 km, effective radius=10 µm). The ratio between the spectral irradiance simulated for cloudless and cloudy conditions is presented in Figures S1 and S2.

[Figure]

**Figure S1: Ratio between the surface solar spectral irradiance at 305 – 325 nm reaching the Earth surface with and without low altitude clouds.**

[Figure]

**Figure S2: Ratio between the surface solar spectral irradiance at 305 – 325 nm reaching the Earth surface with and without high altitude clouds.**

A Cloud Optical Depth (COD) of 1 was considered for the cirrus cloud (Figure S1), while a COD of 6 was considered for the low altitude cloud (Figure S2). For the low altitude clouds, the spectral shape of the attenuation is similar for the SZAs of 45° and 67° and the dependence from wavelength is in both cases small. For the high altitude clouds the attenuation decreases with wavelength at 45° and increases with wavelength at 67°. Attenuation of the irradiance at wavelengths larger than 320 nm is similar at both SZAs. At wavelengths that are affected stronger by ozone attenuation by high altitude clouds is significantly stronger at 45°. Performing the same analysis for different CODs ranging from 1 to 30 for the two cloud types resulted to similar conclusions. From the above discussion it is evident that changes in the occurrence of cirrus would result to more pronounced effects on the levels of UVB irradiance at smaller SZAs.

**Correlation between GPH at 250 hPa, GPH at 850 hPa and tropopause altitude**

As shown in Figure S3, there is a strong, statistically significant correlation between the GPH at 250 hPa and the tropopause altitude over Aosta, Rome and Lampedusa. As also shown in Figure S1 (panels d- f) GPH at 250 hPa is strongly correlated with GPH at 850 hPa.

[Figure]

**Figure S3: Correlation between the absolute anomalies (in m) between GPH at 250 hPa and tropopause altitude for (a) Aosta, (b) Rome, and (c) Lampedusa, and correlation between the GPH at 250 hPa and 850 hPa   for (d) Aosta, (e) Rome, and (f) Lampedusa. The correlation coefficients (cc) are shown at the lower right side of each graph. Values in bold marked with an asterisk denote statistically significant correlation.**

**Ozone trends at different pressure levels**

As shown in Figure S4, ozone over Rome increases significantly at higher stratospheric levels and decreases significantly at lower stratospheric levels.

[Figure]

**Figure S4: Variability and long-term trends (in %) of the ozone mixing ratio at different atmospheric pressure levels over Rome. Average change per year is shown at the lower right of each graph. Values in bold marked with an asterisk denote statistically significant changes.**

**References**

Mayer, B. and Kylling, A.: The libRadtran software package for radiative transfer calculations-description and examples of use, Atmos. Chem. Phys., 5, 1855–1877, 2005.